# Inclusion of People with Intellectual Disabilities in Health Literacy: Lessons Learned from Three Participative Projects for Future Initiatives

**DOI:** 10.3390/ijerph17072455

**Published:** 2020-04-03

**Authors:** Änne-Dörte Latteck, Dirk Bruland

**Affiliations:** Bielefeld University of Applied Sciences, Institute of Education and Health Care Research, Department of Nursing and Health, Faculty of Business and Health, 33619 Bielefeld, Germany

**Keywords:** people with intellectual disability, health literacy, empowerment, health inequality, basic research, participatory research, qualitative research

## Abstract

*Background*: People with intellectual disabilities (IDs) constitute a high-risk group in relation to several diseases. Promoting their health literacy (HL) could be highly beneficial in the management of health information and making informed decisions. However, there are varying ranges of cognitive, communication and literacy levels in people with IDs. According to our literature review, a HL concept for this target group has not been adequately conceptualized. *Methods*: To increase knowledge about the target group, adapted HL results from three innovative (research) projects are presented. *Results*: The key factors are: a) target group orientation; b) social context and everyday life; c) individual resources, like communication and literacy levels; d) a multi-modal strategy to strengthen HL; and e) the self-determination and participation of people with IDs. *Conclusions*: The projects illustrate that the HL of people with IDs has been successfully addressed by taking these key factors into account. A target-group-orientated HL concept could affect more than positive health outcomes; it could also empower a high-risk group in relation to health problems. However, to develop successful action concepts and strategies, valid data are crucial. The heterogeneity of people with IDs is one of the biggest challenges in obtaining such data. Future studies will need to face these challenges.

## 1. Introduction

Health literacy (HL) is a broad concept and is defined as “people’s knowledge, motivation and competences to access, understand, appraise, and apply health information in order to make judgements and decisions in everyday life concerning healthcare, disease prevention and health promotion to maintain or improve quality of life during the life course” [1]. HL is increasingly seen as a crucial factor for health-related outcomes, as well as a prerequisite for empowerment [2,3] and as an influencing factor in relation to health equity [4]. HL levels have effects on health behavior, healthcare usage and therefore healthcare costs [4]. However, intellectual disability (ID) is associated with reduced communicative and cognitive abilities, reading and writing skills and self-perception [5], as well as reduced ability to understand new or complex information and to learn and apply new skills [6]. This leads to the following question: To what extent can HL concepts be applicable for people with IDs?

At present, it is assumed that between 1 and 3 percent of the general population have an ID (IQ < 70) (intellectual disability is divided into mild intellectual disability (IQ 50 to 70–85%), moderate intellectual disability (35 to 49–10%), severe intellectual disability (IQ 20 to 34–3 to 4%), and profound intellectual disability (IQ less than 20–1 to 2%) [7]. We focus on mild to moderate intellectual disabilities, because these disabilities characterize the majority of people with IDs. We recognize that, in this way, we exclude people, but research on this heterogeneity group has to start with people with certain skill levels, and other types of intellectual disability can be taken into account in future research) due to genetic conditions, complications during pregnancy or problems during birth, diseases or toxic exposure, and injuries [7]. The Convention on the Rights of Persons with Disabilities Article 25 states: “Health States Parties recognize that persons with disabilities have the right to the enjoyment of the highest attainable standard of health without discrimination on the basis of disability” [8]. However, there is evidence that people with IDs suffer earlier from age-related health problems (especially for chronic illnesses) than the general population, such as a reduced walking ability or a particularly high morbidity rate [9,10]. The life expectancy of those people has risen sharply in recent decades. Hence, they frequently have health limitations, such as incontinence, difficulty swallowing, sensory loss, cardiovascular disease, adaptability losses, oral motor problems, musculoskeletal deformities and cognitive decline [11,12]. Additionally, they receive fewer diagnostic and therapeutic measures, such as preventive and/or screening examinations, and access to prevention and health promotion programs is limited for them [13]. Furthermore, a lack of awareness of unhealthy lifestyles, e.g., unhealthy nutrition, low levels of physical activity and incorrect medicine intake, are likewise associated with this group [13]. Especially regarding these last points, it is assumed that the promotion of HL could be highly beneficial in improving health and quality of life in this population group, but currently, there is no good evidence that it does.

The results from a literature review [14] illustrated that most studies dealing with the HL of people with IDs have a deficit-oriented perspective and, according to Nutbeam’s HL-Typology [15], hardly go beyond a purely functional understanding [16]. In contrast, communicative and social competencies that are relevant to knowledge for the sake of making health-related decisions are rarely addressed in HL interventions for people with IDs. To date, only Chinn [16] has discussed a HL conceptualization for people with IDs. She criticized that often only a functional perspective is taken, so that health-related information is changed into easy-to-read language, and interventions are developed for time-limited and specific group format interventions, but none of these measures address the real everyday needs of people with IDs. While the twelve studies identified in the literature review outline that the level of HL within this group is very low, they fail to invoke scientific evidence to support this claim [14]. In addition, the perspectives of people with IDs are not directly questioned, as in most studies, health professional are asked about HL and report on behalf of this group. A further look shows that the HL of people with IDs is mostly used to describe barriers to health care or to use a theoretical framework, but without target group specific modifications [14]. HL is included neither in the research findings nor in the test variables. It seems that adequate methods to measure HL in people with ID do not exist [14]. Nevertheless, in nearly all of the included studies, the measurement and conceptualization of HL is assumed to be important for designing effective interventions that promote the HL of people with IDs [14].

Research and practical investigations on people with IDs seem to be lacking in the HL research and practice that address the general population or other target groups. People with IDs still seem to be a ‘hidden’ population in HL research [17]. However, to develop successful action concepts and strategies, valid data are crucial, and obtaining such data is complicated, as there is a lack of research on key factors relating to the access and use of information that takes into account the particularities of people with intellectual disabilities [18]. To increase this knowledge, we present two different research projects and describe the best practical example. The advantage of this approach is that it brings together lessons from different sources, with a view to future initiatives that may bridge the research gap. To achieve this, we present different initiatives that involve the analysis or promotion of the HL of people with IDs. Our primary aim is to answer the question regarding to what extent HL concepts are applicable for people with IDs or whether a target group-orientated adaption is necessary. To achieve this, a) we analyze key factors relating to the promotion of the HL of people with IDs by explaining projects with a participatory approach, which seems to be innovative, according to the literature review; and b) we discuss how the HL of people with IDs can be surveyed. As mentioned above, there is a lack of measurement instruments [19] and insights for key factors relating to the access and use of information that takes into account the particularities of people with IDs [18]. One shortcoming is that the findings of the mentioned projects run side by side but have not yet been brought together. This represents a first step towards theoretically bundling research projects and deriving initial findings from them. Hence, the lessons from the presented projects provide important knowledge on the HL of people with IDs.

## 2. Methods

Using a qualitative content analysis [20], protocols from group discussions in initial events of the research project, ‘with enthusiasm and energy throughout the day’, were reanalyzed according to the questions regarding the key factors relating to the promotion of the HL of people with IDs. In a second step, the results of our two research projects were compared to find the common key factors for promoting HL. The project, ‘Health Literacy in older people with intellectual disabilities’, already examines the significance of the health literacy concept for older people with IDs, so the results were used without re-analysis. For comparison, we screened the results of both projects in terms of the common key factors. Based on the two research projects, we also discuss how the HL of people with IDs can be surveyed. Furthermore, the outcomes were compared with the best practical example, and the main aspects for further research were noted. At last, a final discussion in our institute was performed and noted. This was an exploratory approach, without standardized research methods, using existing data, with the aim of generating insights in this field. However, the advantage of this approach is that it brings together lessons from different sources in a field with little available evidence, and it has a benefit for future initiatives relating to the promotion of the HL of people with IDs. In this way we like to share our experiences for future work and debate.

## 3. Health Literacy Research Projects and the Best Practical Initiative Example

For each project, we describe the background and then provide insights into the project approach and the lessons learned that are relevant for HL. The focus is on various aspects. Each project ends with implications regarding HL, according to the findings, which are summed up at the end of the chapter.

All presented projects (Table 1) have a participatory approach and take the perspective of the people with IDs themselves. Two presented research projects are conducted in our institute. The best practical project is, to our knowledge, the longest continuously running project relating to the promotion of the HL of people with IDs. For more details, especially regarding the research method, information about previous publications are given or it will be published later.

### 3.1. Health Literacy of Older People with Intellectual Disabilities (Unless Otherwise Specified, All Information is Based On *[18,19,21]*).

#### 3.1.1. Background

In order to develop a target group-specific HL concept in further research, knowledge on how to analyze the perspective of people with IDs and indicate the determinants of a target group-specific HL concept must first be generated. Elderly people with IDs constitute an appropriate group due to the fact that health-related topics are more important in their life. The increasing lifespan of people with IDs is associated with an increase in age-typical morbidities, such as dementia, physical and mental fatigue and reduced attention, but also chronic diseases, such as diabetes mellitus or cardiovascular disease [11,19,21].

#### 3.1.2. Approach and Lessons Learned

In this project, a total of 31 interviews were conducted with people with IDs themselves (mean length of 17 min). The interviews were supported by guidelines. The basic dimensions of these guidelines were related to the definition of HL of Sörensen et al. [1]: (a) access to health-related information; (b) implementation and application of health-related information; (c) communication behavior on health issues; (d) design of a self-determined decision; (e) health and disease experience; and (f) future visions, fears, and challenges. With regard to the participants’ answers to the questions, how older people with IDs understand their health and illness and what consequences result from this that are relevant to the HL concept are analyzed. The approach was approved by the independent ethics committee of the University of Bielefeld, and included verification of the informed consent of the respondents and the legal representative by means of target group-specific information. Specific data collection conditions had to be respected. The participants were aged 50 and over (oldest: 81 years old; mean: 61 years old) and lived in an institution for people with IDs. Sufficient German language skills were a prerequisite for participation. Expected were the mentioned reduced communicative and cognitive skills, short interview times and a change from narrative communication to a question-answer form due to lack of attention or concentration. The validity of the responses of people with IDs during the interviews is threatened by a number of biases, e.g., social desirability, with a tendency to answer "yes" to questions [22]. It is suspected that this is related to institutionalization and power asymmetry, which is countered by a positive response behavior [23]. Due to the heterogeneity in cognitive and verbal abilities within the target group, phase-dynamic interviews [24] were conducted, which included both narrative- and problem-oriented aspects. The advantage of this type of interview is that its progress can be flexibly adapted to the situational and communicative competences of the interviewee and the current constitution of attention. Furthermore, the questions mainly had an open form to avoid social desirability answers.

People who belong to a particular collective often share the same horizon of experience and use a shortened language in interviews with people from outside that collective. This language is recognized and accepted within the collective, but it makes data evaluation difficult due to the implicit codes and abbreviations [24]. For evaluation, a specific analysis method was chosen, called the integrative basic method. It was very important that this method particularly emphasizes the principle of openness towards the persons, situations and methods. At the same time, it emphasizes the high relevance of a process of understanding others, and the selection and formulation of meaning and relevance is only determined by the participants and not in advance by researchers [24].

#### 3.1.3. Key Results and Implications

This study confirmed that the perspectives of people with IDs concerning HL can be surveyed using adequate methods. The handling of health-related information depends on the area in which the interviewee is currently active. Therefore, the interviewed people divide health and illness into three different areas, which are strongly separated from each other. These are healthcare, disease prevention and health promotion. In these different areas, the perception of one’s own power to act has a decisive role. Additionally, older people with IDs attribute these three areas to certain determinants. For example, they characterized healthcare as taking medicine and continuing to rest (e.g., from the interview, “Then I’ll go to the doctor and he’ll fix it.” INT.8 [21]) and health promotion as preventive medicine examinations (e.g., “Precautions are made to keep you healthy” INT 7 [21]) or the usual (daily) routine (e.g., from the interview, “Health—this involves having a warm bed cover and wearing really warm clothes in winter, so that one does not freeze. You should be healthy—comb your hair, clean your ears, and wash your ears” INT.5 [21]). This also has an impact on people’s motivation within the different areas to make health-related decisions for themselves or to let others make them. The interviews showed that a deficit-oriented or functional perspective does not lead to the promotion of HL. If the process of informed self-determination is to be supported in the HL concept, it is necessary to include the resources and skills of people with IDs. Therefore, individual ways of addressing these HL resources and abilities must be possible. The following aspects shall be taken into account in measuring the HL of this target group: (a) orientation to living conditions; (b) motivation to maintain the ability to work (which could be used for prevention); (c) own power to act; and d) condition and action components (IF THEN) (which should be used to provide information).

### 3.2. “With Enthusiasm and Energy throughout the Day”—How can Perspectives of People with ID be Taken into Account? (Basic Information according to *[25,26]*).

#### 3.2.1. Background

This project focuses on users’ perspectives and how people with IDs understand, appraise, and apply health information relating to physical activity in order to develop and implement strategies in daily life to promote physical activity (PA). While systematic reviews indicate that people with IDs show significantly lower levels of physical activity than the general population and often lead sedentary lifestyles [27,28], they are mostly unconsidered in concepts in relation to the promotion of PA [29]. By promoting PA, the risk of disease can be minimized, existing diseases can be managed better, and the course of a disease can be positively influenced [30,31]. In this project, the model of physical activity-related health competence [32] was used, with its subcategories: movement competence, control competence and PA-related self-regulation (motivational- volitional). This model contains health literacy mainly in the subcategory of control competence, which includes the body and physical-related basic knowledge in relation to health [32].

#### 3.2.2. Approach and Lessons Learned

We used a participatory research approach to include the perspectives of people with IDs. In this way, we can link health promotion to self-perceived living conditions, as well as the relevant power to act for oneself. From the start of the project, an expert board, including four members, two of whom had an ID, was established. In this group, all relevant project steps were discussed, for example, the title of the project or how to inform all people in the institution. The chosen title, “With enthusiasm and energy throughout the day—connecting health with useful activities”, soon became a catch phrase for the whole institution (including the staff). The discussion of this group is noted in the protocols of each session. To inform all people in this institution about this project, initial events in every residence were conducted. The two members with ID explained their visions on how to share project and health information. To achieve this, a quiz was used with a wheel of fortune. In the residences, we told the participants who we are and what the benefit are of being physically active, with a presentation in an easy-to-understand language (2 min), and after that, the quiz started. People could turn the wheel of fortune, which stops on a picture relating to physical activity. This turns into a discussion within the group of people with IDs about health-related topics, like physical activity, sports, nutrition and health advice from health professionals, like a general practitioner (GP). Two examples of the discussions are as follows:

“If you do more exercise, you are allowed to eat more” (GP3). “The doctor says I should be more active by walking, but I have to take the bus to work” (GP1).

The discussions were noted in a protocol. After initial events, we established a research group to prepare our interviews. With this group, existing knowledge on people with IDs was taken into account for the ongoing research project. The group uncovered relevant topics regarding physical activity for the target group. The meeting was noted in a protocol. Based on this meeting, the researcher (author DB) developed questionnaire guidelines, with support from the research group. In this way, the interview guidelines were based on existing health knowledge and experiences and strategies relating to physical activity. Furthermore, research strategies, like observation of a working day, were added. These were not included in our original research strategy, because we assumed that people would not like them. However, through discussion with the research working group, it was requested that the interviewers “actually be present and not just talk about it; how else could you experience our everyday life?” (quote from the research group). In retrospect, the interview guidelines, with topics relevant to daily life, and supportive materials (pictures, little statues about the body, etc.) produced by the research working group worked very well. The intervention is still to be developed, and based on our good experience, it will be critically discussed by the expert board and the research working group. The approach was approved by the independent ethics committee of the University of Bielefeld, and included verification of the informed consent of the respondents and the legal representative by means of target group-specific information.

#### 3.2.3. Key Results and Implications

Our data analysis shows that people with IDs have a significant amount of knowledge about their own health. For example, they know a lot about the relationships between diet and exercise. They also know what good physical activity is, as well as where they have existing physical limitations and what kinds of exercise could have negative consequences for their health. Caregivers, like nurses, who are approached as needed, constitute professional resource for HL. They discuss the instructions of physicians, take a critical view of health advice, if they do not feel understood, and their own health decisions are based on their own experience (adherence). The transformation from health knowledge into action, as ideas for physical activities and possibilities, like training courses, which take the special needs of people with IDs into account, is often lacking. The findings of our research project indicate that the HL of people with IDs go beyond a functional level; communicative and critical skills can be addressed as well. However, the resources for health, like specific skills and life situation, especially according to experience with self-determination, are very different.

### 3.3. “Gesund Sein” (Being Healthy) as the Best Practical Example of Strengthening the HL of People with ID

#### 3.3.1. Background

For people with IDs, it is especially difficult to make good decisions in terms of their health and find adequate health services for their individual health issues. Interventions to promote the HL of people with IDs are very rare. Because there was no training course on promoting the HL of people with IDs, the Wiener Gesundheitsförderung (WIG—Vienna’s Health Promotion) developed a course, with a participatory approach, called “Being Healthy”. The aims of the training course are to:raise awareness of the determinants of health,strengthen the role of experts for one’s own health,promote decision-making and problem-solving skills with regard to health,strengthen people’s ability to use health and social services for their own needs,promote self-determination.

To achieve these aims, relevant knowledge, with a suitable content for daily use, was transferred into an understandable format in a motivating way. The course has now been running successfully for several years. However, this is the first time that such a course has been offered in the international community, and it therefore represents the best practical example (all information was provided by WIG. Information are also from online sources (only in German language) [33,34,35]).

#### 3.3.2. Approach and Lessons Learned

The development of this course was conducted with experts of the alliance of Self-Determinate Life Movement in Austria and in cooperation with a research center. Self-determination is a core element of the course. According to the Self Determinate Life Movement in Austria, self-determination is defined as having control over one’s own life and making all decisions for oneself. This also means that a disabled person is the sole expert in his or her life. The conception and development took nine months (June 2014 until May 2015), which also included a needs assessment of potential participants. Based on the results, the main themes and framework of the course were developed. From May to July 2015, eight courses were offered in a pilot phase to validate the course. To achieve this, after each module, the trainers filled out special developed documentation forms. Additionally, workshops with participants and trainers were conducted. The course was modified based on the feedback from the workshops. Finally, ‘Being Healthy’ consists of six modules, each with a scope of 4 h every week, with a maximum of six participants. The modules are: (1) My daily life; (2) My body and I; (3) Well-being; (4) Ill-being; (5) Sexualities; and (6) The healthcare system and how to use it. The modules include specially developed exercises, so that every participant can link their individual life situation to the course. To achieve this, there are several methods, like role play, movement and sensory exercises. In the modules, there is also space for important cross-cutting issues, like ‘What do I want or what do I not want?’, ‘the bad side of self-determination’, ‘my own opinion—What is right for me?’, ‘the double look (I can’t, but I have to)’, and ‘themes of diversity’. During the course, a personal health folder is designed by the participants themselves. Course trainers receive instructions, including learning goals, a description of the individual components (how to use them), and exercise sheets. To meet the needs of the participants and their self-determination, the following requirements were placed on trainers: they were already trained as an adult educator, and they have high self-reflection and health promotion competences, as well as gender and diversity competences.

Training courses are offered in different spaces, according to the needs of the participants. The centerpiece of the self-determination approach is “free courses”, which take place in the self-representation center in Vienna, a cooperation partner of this project. Interested people register independently for this course or with the support of relatives or caregivers. Typically, participants come to the course alone (but if necessary, with supporters). The courses are advertised with flyers in an easy-to-understand language and with a video clip (see (https://www.youtube.com/watch?v=Eb0hQz4mZBg&feature=youtu.be). There are between two and four free courses per year. There is also the possibility of offering courses in institutions. In order to facilitate decision-making, an information forum is offered in the form of a two-hour information event at the interested institution, in which both people with IDs and their caregivers can participate. The institution must know what to expect and how to support people with IDs, e.g., to keep barriers preventing participating to a minimum. As such, the course is free of charge and should be accepted as educational leave (because people with IDs have only a limited number of days of holiday). Above that, there are two peer groups for people with IDs, because regular exchange of experiences of all participants is very helpful. Meetings are monthly and guided.

#### 3.3.3. Key Results and Implications

In the beginning of 2016, the course was evaluated. Twenty courses were conducted, and the participants and trainers were asked about the course. To achieve this, interviews were conducted personally at the beginning, directly after and 2 months after the course. The essential results were as follows: Two of three people said they would continue to use the health kit, and it was confirmed that more than a half of the participants remembered the course contents for a long time. A statement from a participant was “[I like] the ability to exchange one’s own ideas; that one’s own opinion could be expressed; to be allowed to reveal one’s own knowledge; to cooperate with other course participants”. A further result was the importance of the trainers’ competence. The typical challenges mentioned by the trainers were the heterogeneity of the participant group, intensive preparation and follow-up, and the binding nature of participation. However, the overall results were very positive, and the course design, content and materials are suitable for the participants. Thus, this training course leads to an individual pathway through the health care system, which is tailored to the needs of people with IDs and strengthens their decision-making process in relation to their health. This course is the best practical example of a successful HL intervention in people with IDs. It illustrates the following:the importance of course materials developed for the needs of people with IDs,that addressing the self-determination of people with IDs is possible, and they have the (health) resources to allow for this,that trainers are an important resource,the necessity of high requirements for trainers to address the heterogeneity of the group and to address individual resources adequately,that promoting the HL of people with IDs can be sustainable.

### 3.4. Key Factors of the Presented Projects

The results of both projects were screened in terms of the key factors and were compared with results from the best practical example. In all projects, the key factors of success are (a) the target group orientation; (b) social context and everyday life routines; (c) individual resources, like communication and literacy levels; and (d) a multi-modal strategy for the promotion of HL. In contrast to our findings in the literature review [14], where studies were mainly from the health professional perspective, the presented projects rely on (e) the self-determination and participation of people with IDs during the intervention development and implementation phase.

## 4. Discussion

At the beginning, it was highlighted that, currently, little is known about the HL of people with IDs. HL interventions often do not go beyond a purely functional understanding, and people with IDs are mostly not directly involved in the assessment [14]. People with IDs still seem to be a ‘hidden’ population in HL research [17]. However, to develop successful action concepts and strategies, knowledge and valid data are crucial. To obtain such knowledge and data, three projects are presented to (a) analyze key factors relating to the promotion of the HL of people with IDs, and (b) to discuss how the HL of people with IDs can be surveyed. This chapter presents the results of the final discussion, according to the lessons learned, with a view to determining what future research is required. 

### 4.1. Key Factors relating to the Promotion of the HL of People with IDs

With respect to HL, which is understood as dealing with health information, and in order to make health decisions in everyday life, there are some challenges for this group. For example, people with ID may process information slowly, have difficulties with abstract ideas, and have varying ranges of communication and literacy levels [36]. All three presented projects deal with these so-called challenges to conceptualize and promote the HL of people with IDs using the mentioned key factors. It was possible to take into account the communicative and critical skills of the target group (see project: ‘With enthusiasm and energy throughout the day’). The successful key factors take the heterogeneity of abilities into account, e.g., different writing and communicative skills and very different health experiences. On the other hand, this illustrates the main challenges, mainly relating to finding generalizable relevant determinants of HL in this heterogeneity group.

### 4.2. Surveying the HL of People with IDs

It was often assumed that the HL of people with IDs is very low, without naming a reference source for this statement [14]. Against this, the lessons that have recently been learned include that it is possible to measure the HL of this group. In all presented projects, this was conducted via a qualitative research design. This was the preferred way to respond to the very different abilities and challenges associated with surveying people with IDs (project: ‘Health Literacy in older people with intellectual disabilities’).

### 4.3. Do We Need A Target Group-Orientated Adaption?

People with IDs have the right to access information in an understandable way and to be supported in finding adequate health support that grants them autonomy, scope of action, etc. HL could promote participation in health-related decisions (empowerment). Up to now, there seems to be no adequate HL concept that considers the specific personal, situational and societal determinants of people with IDs. In all projects, the HL definition is used and adapted to the demands of the participants as described, but the specific conditions of this group have not been provided in the HL debate to date [14]. Specific HL concepts offer great advantages, like drawing attention to a neglected area of research, meeting the specific needs of the target population and leading to domain-specific assessments [37]. Based on the experience and results of the presented projects, it was illustrated that a target group-orientated concept of HL could affect more than positive health outcomes; it could also empower a high-risk group. Therefore, a target group-adapted HL concept should be developed and compared with the existing conceptual models. Existing and future projects have to provide new scientific knowledge and insights that can be used to adapt HL for people with IDs.

### 4.4. Future Directives and Challenges

The generalization of our findings is limited, mainly due to the fact that the research was conducted in specific contexts. Further research is needed and the existing research needs to be brought together to raise knowledge in this field and make people with IDs visible in HL research. An important future task is to bundle more results of other projects in this field, also from other countries, in a standardized way. These include, for example, the project, ‘Health Promotion with People with Disabilities’ [38], and the project, ‘Drug Management and Health Care for People with ID’ [39], both of which have a participatory approach to the promotion of health and evaluation of health skills. Both projects are still running, and the HL issues have to be discussed in detail. The lessons learned from the three presented projects can be taken as a basis for further research projects.

It has to be stated that research on people with IDs has qualitatively high demands in terms of the development of adequate formats for data assessment, such as an understandable language, and thus an additional expenditure of (time, material and personnel) resources. Moreover, there are further demands on researchers, for example, people with ID often express their wishes and needs in a way that cannot be clearly interpreted. Their behavior is irritating, perceived as not conforming to standards and therefore easily misinterpreted. Researchers in the field are unexpectedly confronted with existential borderline situations in the company of this extremely vulnerable group of people, which is experienced as the limit of their own research role. The field of conditions requires researchers to have a recognizable basic attitude, which is characterized by a high degree of sensitivity, ethical reflection and self-reflection [40]. Because the social context and circumstances in the life of people with IDs is too multidimensional and too individual for them to be captured by one discipline alone, this has to be conducted in a transdisciplinary way (e.g., special pedagogy, nursing science, social pedagogy, etc.) [38]. Practitioners and caregivers, like nurses or social workers, are a professional health literacy resource, while they support people with IDs in their daily life routines and situations. Therefore, they have to be taken into account, while maintaining the self-determination of people with IDs. For example, they could help by breaking down communication barriers in daily life, supporting the completion of survey instruments or participating in interviews for research [40]. An adapted HL concept for people with IDs should include programs for the promotion of the communication of caregivers according to HL.

Qualitative research methods might be better able to understand the health literacy of people with IDs. However, a larger research cohort is needed to obtain valid and reliable results, and quantitative methods are needed for this. In this way, the knowledge already gained can be substantiated. However, general problems, such as the low reading and writing skills in this group, often result in incomplete outcome data [29]. Next to this, there is also the learned social desirability of questions, with an extreme response behavior of the respondents. If the participants fill out a self-assessment questionnaire, which is mainly used for HL measurement in the population, very positive answers can be expected (ceiling effect) (see description in the project, ‘Health Literacy in older people with intellectual disabilities’). Therefore, a survey using common measurement instruments in this target group seems to be not useful. A third-party assessment does not do justice to people and their right to self-determination. An adequate quantitative measurement tool to assess HL in people with IDs is non-existent [14]. For future research, these findings are beneficial, e.g., a new project at our institute involves promoting the HL of people with IDs using modern techniques, like offers on the internet. Online videos could be used as a medium for the dissemination of information. In a future project, the content will be developed using a participative research approach, and videos could be prepared under learning theoretical concepts. The aim of this project would be to test and validate an adequate measuring instrument. To this end, the results of the projects presented in this study are a great asset for further work in the field of the HL of people with IDs.

## 5. Conclusions

People with IDs still seem to constitute a ‘hidden’ population in HL research [17]. The presented projects illustrate that people with IDs have special limitations and resources in relation to HL, but it is possible to promote and analyze HL in this group. Communication and critical skills can be addressed using adequate methods. For future activities, the lessons learned from our projects can be incorporated into theoretical considerations and concepts of HL for people with IDs. It is assumed, from these projects, that a higher HL would help healthcare professionals to communicate with people with IDs. Besides the expected outcome parameters for health, a higher quality of life is expected for people with IDs. It can be assumed that the promotion of HL in people with IDs may havea positive impact on people with IDs themselves, their (family) caregivers and the healthcare system. Therefore, a target group-adapted HL concept is needed. The lessons learned indicate that a target-adapted concept of HL could affect more than positive health outcomes; it could also empower a vulnerable group in terms of health problems and support people with IDs by promoting their participation in health-related decisions, and this will reduce health inequalities.

## Figures and Tables

**Table 1 ijerph-17-02455-t001:** Overview of the described projects.

Project Title	Health Literacy in Older People with Intellectual Disabilities	With Enthusiasm and Energy throughout the Day	Being Healthy
**Aim**	Research project to identify the determinants of a target group-specific HL concept	Research project with a participatory approach to develop a target group-oriented intervention in order to promote a physically active lifestyle in people with IDs	A developed training course to promote the HL of people with IDs
**Objective**	Analyzing the perspective of older people with IDs on health topics using adequate survey and evaluation methods	Participatory research approach to survey the perspective of people with IDs on health-related topics and to uncover existing health resources and competencies	The course leads to an individual pathway through the health care system, tailored to the needs of people with IDs
**Country/Duration**	Germany: 2012–2016	Germany: 2018–2021	Austria: since 2012

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
