# Peer review of "Inclusion of People with Intellectual Disabilities in Health Literacy: Lessons Learned from Three Participative Projects for Future Initiatives"

_ijerph, 2020, doi:10.3390/ijerph17072455_

Round 1

Reviewer 1 Report

Overall: This is an interesting report of 3 health literacy studies/interventions with adults with intellectual disabilities. Health literacy is an important topic in public health, and the population with intellectual disabilities certainly needs special consideration for health care and health literacy. As interested as I was in the topic, however, this paper was very difficult to review - At over 6,000 words, it is much too long; and the writing is often unclear, ungrammatical and repetitive – the key information could be conveyed more effectively in one-half the length. I would recommend major revision.

Title:  It would be clearer to state that this paper describes lessons learned from 3 programs.

Introduction:  Line 58: “Above that, a lack of awareness of unhealthy lifestyles are likewise associated with this group [13].” – Specify what is meant by “unhealthy lifestyle” – are people with ID more likely to have unhealthy diets or limited physical activity? Or are they heavily using tobacco, alcohol and other drugs?

Health Literacy Research Projects:

This section needs a description of the Methods used: Are these 3 studies that the authors conducted, as suggested by the statement in line 168, “We used a participatory research approach to include the perspectives of people with ID,”? If so, there needs to be detailed description of human subjects protocol approval and the process of informed consent for people with ID.  Or were these 3 studies conducted by others and identified by a literature review that used certain search terms to search certain databases, and identified a certain number of articles that met certain criteria for inclusion…?

Lines 108-109: “question how older people with ID construct their health and illness?” – their self-concept of health and illness? “And, what consequences result from the answers to this question to the HL concept?”—clarify what this refers to.

Lines 206-207: “From the interviews we can conclude, that people with ID know exactly what their state of health is as well as what they are (not) allowed to do” – This seems to be an over-generalization. Even people without ID often do not know exactly what their state of health is, what they should and shouldn’t do.

Lines 132-135: Clarify this.

Discussion: At nearly 2,000 words, this is far too long, and could be cut by one-half or more. Much of it repeats what was said in the earlier sections Introduction and Results. This section should briefly summarize findings, compare to other studies, and give recommendations for future research and interventions.

Conclusion: This is also far too long and repeats what was said in Results and Discussion. One brief paragraph would be best.

Author Response

Thank you very much for your very detailed feedback and your time to review our article. Your view was important for us to see where are problems of understanding, which seems a major problem. We edit our article mainly of these understanding and hope, that our approach and aim is much clearer now. In general, the whole text is proofed to language, correct comma and formatting due to your feedback. Overall, your comments are very valuable for us and we edited our article (see in table).

Reviewer 1

This is an interesting report of 3 health literacy studies/interventions with adults with intellectual disabilities. Health literacy is an important topic in public health, and the population with intellectual disabilities certainly needs special consideration for health care and health literacy. As interested as I was in the topic …

Thank you for your positive response to our topic, which is a motivation for us.

At over 6,000 words, it is much too long; and the writing is often unclear, ungrammatical and repetitive – the key information could be conveyed more effectively in one-half the length.

We edit the chapter and shortened the article.

Title:  It would be clearer to state that this paper describes lessons learned from 3 programs.

Thank you for this hint. We changed our title due your comment.

Introduction:  Line 58: “Above that, a lack of awareness of unhealthy lifestyles are likewise associated with this group [13].” – Specify what is meant by “unhealthy lifestyle” – are people with ID more likely to have unhealthy diets or limited physical activity? Or are they heavily using tobacco, alcohol and other drugs?

There are now examples that specified this statement.

This section needs a description of the Methods used

We added a method chapter. We hope, we can better communicate our approach.

: Are these 3 studies that the authors conducted, as suggested by the statement in line 168, “We used a participatory research approach to include the perspectives of people with ID,”? If so, there needs to be detailed description of human subjects protocol approval and the process of informed consent for people with ID.  

An ethical statement is added in the projects.

.  Or were these 3 studies conducted by others and identified by a literature review that used certain search terms to search certain databases, and identified a certain number of articles that met certain criteria for inclusion…?

Thank you very much. We have taken your comment as an opportunity to describe the starting point of the article more clearly. A description is added.

Lines 108-109: “question how older people with ID construct their health and illness?” – their self-concept of health and illness? “And, what consequences result from the answers to this question to the HL concept?”—clarify what this refers to.

We edit the sentence and edit new information, so that this should be clearer.

Lines 132-135: Clarify this.

We clarify this part with a focus on the approach of this article.

Discussion: At nearly 2,000 words, this is far too long, and could be cut by one-half or more. Much of it repeats what was said in the earlier sections Introduction and Results. This section should briefly summarize findings, compare to other studies, and give recommendations for future research and interventions.

We edit the chapter and shortened the article.

Conclusion: This is also far too long and repeats what was said in Results and Discussion. One brief paragraph would be best.

We edit the chapter and shortened the article.

Reviewer 2 Report

Dear authors

Thank for submitting your research paper entitled “Inclusion of people with intellectual disabilities in Health Literacy. Current state of research, present initiatives and future activities” to IJERPH, that is in line with previous study in this journal:

Geukes, C., Bröder, J., & Latteck, Ä. D. (2019). Health Literacy and People with Intellectual Disabilities: What We Know, What We Do Not Know, and What We Need: A Theoretical Discourse. International Journal of Environmental Research and Public Health16(3), 463.

While the research topic is interesting, the paper is difficult to follow due to it is a sum of conclusions from different projects, with a mix of research methodologies. Please, see below my specific comments:

Line 28: you should explain the subject for that heterogeneity: e.g. people with ID? Projects? I suggest using other keywords different than those included in the title for a better paper´s indexation. In addition, “qualitative research” or “basic research” seems too general terms. I suggest including a clear definition of HL at the beginning of the introduction. Line 44. “people with” looks incomplete. g. Lines 52 (9-10) and 55 (11 – 12) is an example of a lack of consistency for the format of the reference. Lines 67-70. This sentence is confused because you are talking about “specific” settings and “real needs” for people with ID. In which context are you talking about? (i.e., community, school, healthcare…) Line 86. When you introduce the projects, I am wondering to know the criteria for determining that are “innovative”. In addition, it is pertinent to define the inclusion criteria for the projects included in Table 1. Section 2.1.3. I miss a definition of the theoretical framework used for that study/project and some of the findings can be improved using qualitative sentences/testimonies. Line 155. Insert a blank space before the number between brackets. Line 160. How do you determine (or measure) the “lower levels of physical activity”? Line 182. GP abbreviation should be explained. Line 184. What “DB” abbreviation means? Lines 184-186. What type of “existing knowledge”? I miss references or a statement that this an author’s novel contribution. Line 187. Remove comma before “were added” Line 196. Beginning a sentence with “Our experience shows…” provide a lack of rigor on the manuscript. Conclusions must come from the evidence, and this is difficult to follow across the manuscript due to the differences between projects and the methodologies used. Line 203. What type of motivation (e.g. intrinsic, extrinsic)? Again, the lack of theoretical frameworks is a weak point for this paper. Line 206. Remove comma after “conclude” Line 211. I suggest including some references when you are talking about “general population” (i.e. for comparison) Lines 212-214. Three times HL in the same sentence makes difficult it's understanding. Lines 233-234. References and/or a theoretical framework about self-determination is welcome. Lines 292-293. You should include the references for the previous publications regarding the projects included in this study. Line 310. Insert a comma before “d)” Line 318-319. Again, references for the general population are required. Line 323. Remove comma after “illustrated” Line 333. “projects” instead of “project” Lines 337-339. Have you really measured the impact of the HL?    Line 251. How much time do you require to train an adult educator? Point 2.2.3. In general, it seems that the external validity of this study is limited. Line 347. Insert “.” Before “Above” Lines 344-353. The information included in this section of the paragraph is not clearly related to the aims of this paper. You are talking about the general limitation of researching with people with ID. In that case, if you had those limitations, how much do you think your results could be biased by these constraints? Lines 366-367. This is the third time that you are comparing with the “general population” but any reference is included. Line 371. HLS.EU should be explained or a reference should be included. Line 389. There are other training programs for people with ID and the discussion is only based on the findings/experiences from the “Being Healthy” program. Line 393. Which type of programs have you planned? Which suggestions or prospective can you provided from your findings? Line 395. Review inverted comma. Conclusions section. Many of the information in this subsection is repeated from previous paragraphs of the manuscript. This makes redundant the reading. Line 371. UN-CRPD should be explained or a reference should be included. REFERENCES: there are many format mistakes such as upper cases of the titles, italics in the journal´s title, the abbreviation of the journal´s title, uniformity of the page numbers, letter size (e.g. Ref. #29)….In addition, I suggest that references in German should include a translation into English.

Author Response

Statement Reviewer 2

that is in line with previous study in this journal:

Geukes, C., Bröder, J., & Latteck, Ä. D. (2019). Health Literacy and People with Intellectual Disabilities: What We Know, What We Do Not Know, and What We Need: A Theoretical Discourse. International Journal of Environmental Research and Public Health16(3), 463.

Thank you for recognizing our previous work, that is included in our article (project 1) and thought one step further in combination with the other projects.

While the research topic is interesting, the paper is difficult to follow due to it is a sum of conclusions from different projects, with a mix of research methodologies

Thank you very much for the analysis. We have put the results of the projects in the foreground of the article. Our approach is now described in detail. We hope that this gives a better understanding to our results. We have formulated the results more clearly and to remove repetitions.

Line 28: you should explain the subject for that heterogeneity: e.g. people with ID? Projects?

We edit the abstract.

I suggest including a clear definition of HL at the beginning of the introduction.

Now the introduction starts with the definition of Sörensen et al. (2012).

We are not sure, what do you mean with a “clear definition of HL” because the definition of Sörensen (2012) is often used for HL research. We discussed this point and we refer in the article to the definition more to health literacy and think, the article becomes much clearer.

Line 44. “people with” looks incomplete.

That’s right, we complete it.

Lines 52 (9-10) and 55 (11 – 12) is an example of a lack of consistency for the format of the reference.

We proofed and edit it.

Lines 67-70. This sentence is confused because you are talking about “specific” settings and “real needs” for people with ID. In which context are you talking about? (i.e., community, school, healthcare…)

Thanks for this hint. We changed the sentence and guess, it is now better understandable.

“To this date only Chinn [17] discusses a HL conceptualization for people with ID. She criticizes that based on a functional perspective, health-related information is changed into easy-to-read language and interventions are developed for time-limited group format, but none of these measures address the real needs in everyday life for people with ID “

Line 86. When you introduce the projects, I am wondering to know the criteria for determining that are “innovative”.

We establish a connection to the statement how we meant it – “– which seems to be innovative according to the literature review”.

In addition, it is pertinent to define the inclusion criteria for the projects included in Table 1.

Now it is mentioned in the text that two of the projects are conducted by our institute and one is a best practice example and the reason, why we take this.

Section 2.1.3. I miss a definition of the theoretical framework used for that study/project

We now mention that the definition of Sörensen et al. (2012) is the theoretical framework for this study.

and some of the findings can be improved using qualitative sentences/testimonies.

durch qualitative Sätze/Zeugenaussagen verbessert.

We discussed this while writing this article and decided against this regarding the word count. We guess, that through editing the article the approach is much clearer.

Line 155. Insert a blank space before the number between brackets.

It is edited.

Line 160. How do you determine (or measure) the “lower levels of physical activity”?

We guess, that this is not the main focus of the article. The quote is a systematic review and the included studies measure the physical activity with motion sensors and questionnaires. This is critical discussed. We added in the text that the information is from systematic reviews and added a review, for people who are interested in. 

Line 182. GP abbreviation should be explained.

The full name general practitioner is added.

Line 184. What “DB” abbreviation means?

We added the word ‘author’ to make it more comprehensible.

Lines 184-186. What type of “existing knowledge”? I miss references or a statement that this an author’s novel contribution.

We edit the sentence, so that is more understandable.

Line 187. Remove comma before “were added”

It is removed.

Line 196. Beginning a sentence with “Our experience shows…” provide a lack of rigor on the manuscript.

We changed that and should be more understandable in relation to the Method chapter inserted.

Conclusions must come from the evidence, and this is difficult to follow across the manuscript due to the differences between projects and the methodologies used.

We added this chapter to make it more understandable.

Line 206. Remove comma after “conclude”

It is removed.

Lines 212-214. Three times HL in the same sentence makes difficult it's understanding.

We fixed it.

Lines 233-234. References and/or a theoretical framework about self-determination is welcome.

Is added.

Lines 292-293. You should include the references for the previous publications regarding the projects included in this study.

We edit the chapter.

Line 310. Insert a comma before “d)”

We insert it.

Line 323. Remove comma after “illustrated”

It is removed.

Line 333. “projects” instead of “project”

We edit it.

Line 251. How much time do you require to train an adult educator?

We edit the sentence for better understanding.

Point 2.2.3. In general, it seems that the external validity of this study is limited. (also lassen sich die Ergebnisse generalisieren)

Now, we have mentioned it in the discussion, this should be much clearer now.

Line 347. Insert “.” Before “Above”

We insert it.

Lines 344-353. The information included in this section of the paragraph is not clearly related to the aims of this paper. You are talking about the general limitation of researching with people with ID. In that case, if you had those limitations, how much do you think your results could be biased by these constraints?

The chapter is edited, this should be much clearer now.

Line 389. There are other training programs for people with ID and the discussion is only based on the findings/experiences from the “Being Healthy” program.

We edit the chapter and we hope that it is much clearer now. 

Line 393. Which type of programs have you planned? Which suggestions or prospective can you provided from your findings?

We edit the chapter and think it is much clearer now.

Line 395. Review inverted comma.

It is fixed.

Conclusions section. Many of the information in this subsection is repeated from previous paragraphs of the manuscript. This makes redundant the reading.

We edit the chapter and think it is much clearer now.

REFERENCES: there are many format mistakes such as upper cases of the titles, italics in the journal´s title, the abbreviation of the journal´s title, uniformity of the page numbers, letter size (e.g. Ref. #29)…. In addition, I suggest that references in German should include a translation into English.

We proofed and fixed it.

Reviewer 3 Report

This is a research project to increase knowledge of promoting the health literacy (HL) for a target people with intellectual disability (ID), which adapted HL results from three innovative (research) projects. Then the authors produced some factors, including 1) a target group orientation 2) regarding the social context and everyday life, 3) taken into account the individual resources like communication and literacy levels 4) using a multi-modal strategy to strengthen HL 5) self-determination and participation of people with ID, to make statement of current researches, present initiatives and future activities.

This manuscript is a project not article. The authors try to make some statements based on 3 projects (table 1); however, these projects are continuous not completely be end. Thus, there were little solid results to use. I would like to suggest some qualified and quantified results from these 3 projects would be helpful for this project.

The Journal (IJERPH) publishes original articles, critical reviews, research notes, and short communications, but not projects nor proposals. “… Authors should not unnecessarily divide their work into several related manuscripts. …” was noted on the instructions for authors.

Author Response

Statement Reviewer 3:

Thank you very much for your critical statement and your time to review our article. However, we have problems to accept most statements for reasons given in the table and hope with our answer, we can increase the understanding of our way writing this article.

Reviewers statement

Answer

The Journal (IJERPH) publishes original articles, critical reviews, research notes, and short communications, but not projects nor proposals. “… Authors should not unnecessarily divide their work into several related manuscripts. …” was noted on the instructions for authors.

We like to show up the way to the special issue, so that it would be understandable why we write our article in this way:

We had to make an abstract about our contribution before we are allowed to submitting our article for this special issue for health literacy. In this abstract we pointed out: “In line with the special issue, we present the research status and practice initiatives towards HL in the context of PwID. Following the current results from the HL research (in PwID), we will demonstrate our research projects to analyze and promote HL in PwID”. The editors accepted this and we stick to the information given in the abstract. Our topic is very valuable because health literacy is an important topic in public health, and the population with intellectual disabilities certainly needs special consideration for health literacy and is still a hidden group (Chinn 2016). We would be sad to put so much time in a valuable article to this topic, and that your point is not stated after submitting our abstract (hence, before we start to write this).

This is a research project to increase knowledge of promoting the health literacy (HL) for a target people with intellectual disability

We cannot agree with that, because in our article we stated clearly that we point out the research gap for this group, present current different projects (in abstract: “we will demonstrate our research projects to analyze and promote HL in PwID, for example”) and based on these results we discuss relevant general determinants for HL in PwID and how this high-risk group can be included in HL research in future.

however, these projects are continuous not completely be end

One of the research project already ended and the other research project (the physical activity project) finished data collection and analysis, which are the main results for this contribution. Furthermore, the best practice example is evaluated and still running for several years successfully, which is a sign of quality for practice projects. We think, that we clearly stated that in our article.

Thus, there were little solid results to use. I would like to suggest some qualified and quantified results from these 3 projects would be helpful for this project.

We edited the results according to the more detailed feedback of the other reviewers.

Thank you very much for your very detailed feedback and your time to review our article. Your view was important for us to see where are problems of understanding, which seems a major problem. We edit our article mainly of these understanding and hope, that our approach and aim is much clearer now. In general, the whole text is proofed to language, correct comma and formatting due to your feedback. Overall, your comments are very valuable for us and we edited our article (see in table).

Round 2

Reviewer 1 Report

The authors appear to have addressed the methodological questions, and most of the suggestions for presenting the results, discussion and conclusion.

I would support publication of this article since it is an important issue, a neglected population, and an innovative intervention and study approach.

The main issue now is that the manuscript must be edited by a professional editor who is a native English speaker/writer. This paper is very difficult to read with improper grammar in every paragraph, and in some places in nearly every sentence. Here are some examples (but there over a hundred similar examples):

219 “The group uncover <uncovered> relevant topics regarding physical activity for the target group.”

238-239: “Caregivers like nurses, who are approached as needed, form an own (professional) resource.” <delete ‘an own’>

244 “From the findings in our research project it could be indicate that..” should be: “The findings of our research project could indicate that…”

267-268 “According to the Self Determinate Life Movement in Austria self-it <??> is defined as having control over the <one’s> own life and making all decisions yourself <oneself>.

269 “Conception and development took nine months (June 2014 until May 2015), which also include <included> a needs assessment”

272-273: "From May to July 2015 eight courses were offered… For this, after each module, trainers fill <filled> out special developed documentation forms.”

350-351" “On the other hand, this illustrate <illustrates> the main challenges, mainly to find generalizable relevant determinants for HL in this heterogeneity <heterogeneous> group."

374-375: “Further research is needed and bring together <??> to raise knowledge in this field and make people with ID visible for HL research”

379: “Both projects <are> still running and HL-relation has <HL issues have> to be discussed in detail.”

419-421: “People with ID still seems <seem> to be a ‘hidden’ population in HL research… The presented projects illustrated, <delete comma> that people with ID have HL <delete this> have special limitations and resources for health, but it is possible to promote and analyze HL in these group <either these groups or this group>.”

426-427: “…people with ID is also to mention <??>. It can be assumed, <no comma> that promotion <promoting> HL in people with ID <add: may> have <add: a> positive impact on people with ID themselves, their (family) caregivers und <and> the health care system.”

Author Response

Again, thank you very much for your feedback and your time. This article was edited by a professional editor. After that, we proofed your examples. We think, the editing has been worthwhile.

The main issue now is that the manuscript must be edited by a professional editor who is a native English speaker/writer. This paper is very difficult to read with improper grammar in every paragraph, and in some places in nearly every sentence. Here are some examples (but there over a hundred similar examples):

This article was edited by a professional editor.

219 “The group uncover <uncovered> relevant topics regarding physical activity for the target group.”

238-239: “Caregivers like nurses, who are approached as needed, form an own (professional) resource.” <delete ‘an own’>

244 “From the findings in our research project it could be indicate that..” should be: “The findings of our research project could indicate that…”

267-268 “According to the Self Determinate Life Movement in Austria self-it <??> is defined as having control over the <one’s> own life and making all decisions yourself <oneself>.

269 “Conception and development took nine months (June 2014 until May 2015), which also include <included> a needs assessment”

272-273: "From May to July 2015 eight courses were offered… For this, after each module, trainers fill <filled> out special developed documentation forms.”

350-351" “On the other hand, this illustrate <illustrates> the main challenges, mainly to find generalizable relevant determinants for HL in this heterogeneity <heterogeneous> group."

374-375: “Further research is needed and bring together <??> to raise knowledge in this field and make people with ID visible for HL research”

379: “Both projects <are> still running and HL-relation has <HL issues have> to be discussed in detail.”

419-421: “People with ID still seems <seem> to be a ‘hidden’ population in HL research… The presented projects illustrated, <delete comma> that people with ID have HL <delete this> have special limitations and resources for health, but it is possible to promote and analyze HL in these group <either these groups or this group>.”

426-427: “…people with ID is also to mention <??>. It can be assumed, <no comma> that promotion <promoting> HL in people with ID <add: may> have <add: a> positive impact on people with ID themselves, their (family) caregivers und <and> the health care system.”

Most given examples were edited by the professional editor. We also checked these examples and two of them were edited by us. Thanks for this examples.

Reviewer 2 Report

Dear authors

Thank you for submitting the revised version of the manuscript (now) entitled "Inclusion of people with intellectual disabilities in Health Literacy. Lessons learned from three participative projects for future initiatives." Although you significantly improve the flow and the understanding of the manuscript, I am still thinking that the contribution to the literature is limited due to i) some overlapping with previous studies by the same research group; ii) the unavailable qualitative testimonies (because of words limit) that would support your findings. 

Sincerely,    

Author Response

Again, thank you for your feedback and your time. We proofed your feedback, see table.

Although you significantly improve the flow and the understanding of the manuscript, …

Thank you very much. Meanwhile, this article was edited by a professional editor. We think, the editing has been worthwhile.

I am still thinking that the contribution to the literature is limited due to i) some overlapping with previous studies by the same research group

The overlapping is true, because the contribution is close to our previous work. To bring together the different projects is an intermediate step. We carried together what we know as a basis for our new project. We made this work as described in this article and are convinced that this knowledge can be very useful for other projects or debates in the field of Hl in people with ID.

; ii) the unavailable qualitative testimonies (because of words limit) that would support your findings. 

We are very grateful for all your feedback on the method. Due to this, we try to make it easier to understand what we have done. We looked for possibilities to strengthen the methods. Now, we show in our projects examples of quotes in support of the claims. It's a step in the right direction. Furthermore, we refer to our publications. (137: For more details, especially regarding the research method, information about previous publications are given or it will be published later.)

We tried to make more, but this takes to many words. In future publications we will describe our results more in detail.  

Reviewer 3 Report

I have no other comments. Remaining as the previous...

Author Response

Reviewer 3

Again, thank you for your feedback and your time. Unfortunately, there were no new comments here. We had already commented your statement. Now, we have edited the article with regard to the feedback of the other reviewers.
